Resistance of cervical vertebrae in response to muscular stresses in pterosaurs: implications for foraging habits and skeletal pneumatization

Buchmann Richard richardbuchmann@gmail.com
Rodrigues Taissa
1 Laboratório de Paleontologia, Departamento de Ciências Biológicas, Universidade Federal do Espírito Santo , Vitória , Espírito Santo , Brazil
2 Programa de Pós-Graduação em Ciências Biológicas, Universidade Federal do Espírito Santo , Vitória , Espírito Santo , Brazil
Hone David
Electronic publication date: 2025 Nov 25
Publication date: 2025
Volume: 13
Electronic Location ID: e20388
Received 2025 Jul 22; Accepted 2025 Oct 24
Copyright: ©2025 Buchmann and Rodrigues
Copyright year: 2025
Copyright holder: Buchmann and Rodrigues
License: This is an open access article distributed under the terms of the Creative Commons Attribution License, which permits unrestricted use, distribution, reproduction and adaptation in any medium and for any purpose provided that it is properly attributed. For attribution, the original author(s), title, publication source (PeerJ) and either DOI or URL of the article must be cited.
License URL: https://creativecommons.org/licenses/by/4.0/

Keywords: Finite element analysis, Biomechanics, Functional anatomy, Cervical column, Pterosauria, Pterodactyloidea

Funding: The Fundação de Amparo à Pesquisa e Inovação do Espírito Santo (FAPES), Brazil #705/2022 The Conselho Nacional de Desenvolvimento Científico e Tecnológico (CNPq), Brazil #314260/2021-8 #178965/2024-3 #406902/2022-4 This study was funded by the Fundação de Amparo à Pesquisa e Inovação do Espírito Santo (FAPES), Brazil, project grant #705/2022 to Taissa Rodrigues and scholarship to Richard Buchmann; and the Conselho Nacional de Desenvolvimento Científico e Tecnológico (CNPq), Brazil, scholarships #314260/2021-8 to Taissa Rodrigues and #178965/2024-3 to Richard Buchmann, and project grant #406902/2022-4 (INCT Paleovert). There was no additional external funding received for this study. The funders had no role in study design, data collection and analysis, decision to publish, or preparation of the manuscript.

==============================
The necks of pterosaurs were flexible and provided mobility for a relatively long skull. The varied morphologies and levels of pneumatization of their cervical vertebrae reflected differences in biomechanical behavior. Here, we examined the structural resistance of the cervical vertebrae to infer the most advantageous movements during the foraging behaviors of two pterodactyloid pterosaurs. We also examined the relationship between vertebral resistance and the presence of pneumatic foramina on the bone cortex. For this purpose, we analyzed three-dimensional models of the cervical vertebrae of Anhanguera piscator and Azhdarcho lancicollis, which are hypothesized to be aquatic and terrestrial predators, respectively, and employed Finite Element Analysis (FEA) to assess and quantify the stresses experienced by the vertebrae due to the performance of six different movement scenarios. We observed that the shorter vertebrae at the ends of the neck of both species favored the proliferation of larger stresses in these regions, especially in the posterior cervicals of Anhanguera piscator and in the atlas-axis of Azhdarcho lancicollis, and that their taller neural arches aided in absorbing stress. Larger stresses at the ends of the neck are consistent with the interior trabecular reinforcement of the atlas-axis and posterior cervical vertebrae, suggesting a link between biomechanical behavior and the level of pneumatization. Additionally, mechanical requirements may have also influenced the presence, size, and number of pneumatic foramina on the vertebral cortex, as evidenced by the large lateral foramen in Anhanguera piscator and the smaller and more numerous ones bordering the neural canal in Azhdarcho lancicollis. Our inferences corroborate the differences in foraging strategies hypothesized for anhanguerids and azhdarchids. The absorption of stresses resulting from ventral pitching of the head and neck indicates that the cervical vertebrae were well-adapted for making rapid movements during predatory hunting. However, variations in the height of the neural spine indicate different mechanical behaviors between these species when raising the skull and neck, which could be faster in Anhanguera piscator while more vigorous in Azhdarcho lancicollis.

Introduction

Pterosaurs were archosaurs, the first vertebrates known to have actively flown (Kellner, 1994; Butler, Barrett & Gower, 2009). As the forelimbs formed their wings, only their heads were responsible for capturing food, as in extant birds (Marek et al., 2021; Marek, 2023). Their similarities can be used to extrapolate functions present in birds to pterosaurs, which can be tested and supported by the phylogenetic proximity between both taxa (Witmer, 1995; Kellner & Campos, 2002; Butler, Barrett & Gower, 2012). Analyses of the functional morphology of bones are also relevant to investigate the performance of previously hypothesized habits of extinct groups (Nachtigall, 1991; Dullemeijer, 2001).

In anhanguerid pterosaurs, their conical teeth were advantageous in catching fish (Kellner & Tomida, 2000; Veldmeijer, Witton & Nieuwland, 2012; Wang et al., 2012; Bestwick et al., 2018; Henderson, 2018; Pêgas, Costa & Kellner, 2020), while the rostral and mandibular crests may have stabilized the beak during predation in water (Veldmeijer, Signore & Bucci, 2007). Together with the mechanical advantage of the jaw adductor muscles, these features show that Anhanguera could have preyed on small to medium-sized fish (Pêgas, Costa & Kellner, 2020). Fish consumption by anhanguerids is also corroborated by isotopic data (Amiot et al., 2010; Tütken & Hone, 2010). On the other hand, azhdarchids have a long rostrum, which suggests generalist habits in terrestrial environments, similar to some modern birds (Prieto, 1998; Witton & Naish, 2008; Witton & Naish, 2013; Padian et al., 2021). However, aquatic habits have also been proposed for Azhdarcho (Averianov, 2013). Therefore, knowledge about the diet of pterosaurs is constantly discussed and tested through morphological studies of dental, rostral, and mandibular elements (Bestwick et al., 2018).

However, cervical mobility must also be considered, as neck movement allows the head to access resources during foraging (Marek et al., 2021; Marek, 2023). In birds, the cervical series is divided into three functional segments, with corresponding anatomical variations in the cervical vertebrae (Boas, 1929; Zusi, 1962; Terray et al., 2020; Buchmann & Rodrigues, 2024a). Furthermore, the avian neck can vary between extremely elongated and short forms, with cervical segmentation playing a fundamental role in the adequate positioning of the head during foraging (Zweers, Bout & Heidweiller, 1994; Molnar et al., 2015). Pterosaurs also present vertebrae that vary anatomically along the neck (Kellner & Tomida, 2000; Bennett, 2001; Bonde & Christiansen, 2003; Averianov, 2010; Eck, Elgin & Frey, 2011; Vila Nova et al., 2015; Buchmann et al., 2017; Andres & Langston Jr, 2021), although forming a less sinuous structure than in birds (Buchmann & Rodrigues, 2024b). Previous analyses indicate that the necks of anhanguerid and azhdarchid pterosaurs had complex ligaments and musculature responsible for the execution of predatory practices (Buchmann & Rodrigues, 2024b; Buchmann & Rodrigues, 2025).

To determine the biomechanical behavior of cervical vertebrae, the muscular forces exerted during movements must be considered (Teng & Herring, 1998; Fastnacht, 2005; Rayfield, 2007; Kupczik, 2008; Porro et al., 2011; Bishop, Cuff & Hutchinson, 2021). Additionally, the influence of postcranial skeletal pneumatization observed in the cervicals of pterosaurs should be examined, as it reduces bone density in the vertebral medullary space (O’Connor, 2006; Claessens, O’Connor & Unwin, 2009; Butler, Barrett & Gower, 2012; Moore, 2020; Buchmann et al., 2021). Their bones are pneumatized via pneumatic foramina, which vary their positions throughout the neck and between species (Wellnhofer, 1991; Kellner & Tomida, 2000; Bennett, 2001; Bonde & Christiansen, 2003; Veldmeijer, Meijer & Signore, 2009; Averianov, 2010; Eck, Elgin & Frey, 2011; Elgin & Frey, 2011; Aires et al., 2014; Vila Nova et al., 2015; Buchmann et al., 2017; Buchmann & Rodrigues, 2019; Andres & Langston Jr, 2021). Due to biomechanical requirements, it is hypothesized that these foramina are distributed in regions of the bone cortex that are less subject to stress (O’Connor, 2004; O’Connor, 2006). Thus, identifying the points most susceptible to stresses caused by cervical movements also has the potential to contribute to our understanding of the distribution of pneumatic foramina along the neck (O’Connor, 2004), and biomechanical analyses that use three-dimensional computational models to apply muscle loads can help to determine the behavior of the studied bones (Porro et al., 2011; Molnar et al., 2015; Kambic, Biewener & Pierce, 2017; Lautenschlager, 2017; Bishop, Cuff & Hutchinson, 2021). Here, we aimed to investigate how stresses were distributed along the neck of two pterosaur species, which allows us to understand the cervical biomechanical behavior during foraging practices, as well as their relation to the arrangement of pneumatic foramina.

Material & Methods

We analyzed cervical vertebrae belonging to Anhanguera piscator and Azhdarcho lancicollis, which are medium-sized pterodactyloid pterosaurs with morphological variations along the neck (Kellner & Tomida, 2000; Averianov, 2010). They were chosen due to the predatory feeding habits hypothesized for anhanguerids and azhdarchids, which are piscivores and terrestrial generalists, respectively (Witton & Naish, 2008; Naish & Witton, 2017; Bestwick et al., 2018; Pêgas, Costa & Kellner, 2020; Buchmann & Rodrigues, 2024b; Buchmann & Rodrigues, 2025). Furthermore, both pterosaurs have pneumatic foramina in different regions in the cervical vertebrae (Kellner & Tomida, 2000; Averianov, 2010).

The analyzed vertebrae of Anhanguera piscator are part of the holotype (NSM-PV 19892) (Kellner & Tomida, 2000), from the Romualdo Formation, in the Santana Group (Araripe Basin), dated to the Albian, Lower Cretaceous, and deposited in the collection of the National Museum of Nature and Science, in Tsukuba, Japan. The specimen has 3D preservation of almost all vertebrae of the neck, except the sixth. Pneumatic foramina are found laterally between the centrum and the neural arch, from the axis to the seventh vertebra. In the eighth and ninth vertebrae, foramina are seen on the sides of the centrum and caudally to the bases of transverse processes.

The cervical vertebrae of NSM-PV 19892 were scanned with computed tomography, which ensured digital preservation of the bone architecture present in the medullary space (Fig. 1A). The tomography scans were performed with a Microfocus X-ray/CT Inspection Systems TXS320 –ACTIS equipment, produced by TESCO Corporation, at 300 to 310 kV and 200 µA, with varying voxel sizes per element (Buchmann & Rodrigues, 2024b). The generated models are available on Morphosource (Boyer et al., 2017), and we provide the DOI of each model separately as Declared Data.

Figure 1 3D models and cross-sections of the analyzed vertebrae.

(A) Anhanguera piscator, eighth cervical vertebra. The 3D discretized model is on the upper center, the sectioned discretized model is on the left side, and an interpretative drawing of the internal pneumatic cavities is on the lower center. (B) Azhdarcho lancicollis, fifth cervical vertebra. 3D model is on the upper right side. The sectioned illustration on the lower right side shows the arrangement of cylinders representing the neural canal and trabeculae in the medullary space, based on Williams et al. (2021). Abbreviations: ic, internal cavities; nc, neural canal; tra, cylinders representing the trabecular bone.

To generate the 3D models of the vertebrae of NSM-PV 19892, we segmented the tomograms using Amira software, version 5.3.3 (Lukeneder, Lukeneder & Weber, 2014). The 3D model corresponding to the sixth cervical vertebra was sculpted from the fifth vertebra using Blender 3D, version 4.0 (Blender Development Team, 2023; Lautenschlager, 2017), with its cortical anatomical characteristics based on the sixth vertebra of the specimen AMNH 22555 (American Museum of Natural History, New York, USA), identified as Anhanguera sp. (Wellnhofer, 1991; Pinheiro & Rodrigues, 2017; Buchmann & Rodrigues, 2024b).

Azhdarcho lancicollis was represented by different cervical elements (ZIN PH 105/44, 131/44, 144/44, 139/44, 147 /44, 138/44, 137/44, 122/44; and CCMGE 1/11915 and 7/11915) (Averianov, 2010) that originated from the Late Cretaceous Bissekty Formation in Dzharakuduk, Uzbekistan, and are deposited in the paleoherpetological collection of the Zoological Institute of the Russian Academy of Sciences (ZIN PH) and Chernyshev’s Central Museum of Geological Exploration (CCMGE), both in Saint Petersburg, Russia. Together, they comprise all 3D preserved cervical vertebrae of Azhdarcho lancicollis (Averianov, 2010). Most azhdarchid vertebrae have pneumatic foramina around the neural canal in cranial and caudal views, except in the atlas-axis (Averianov, 2010). Digital models of each vertebra were generated using a non-contact 3D laser scanning method and provided to the authors by Alexander Averianov (Zoological Institute of the Russian Academy of Sciences, Russia). The generated 3D models are available on Morphosource (Boyer et al., 2017), and we provide the DOI of each model separately as Declared Data.

When necessary, the cortex of the 3D models of the vertebrae of Azhdarcho lancicollis was reconstructed using Blender 3D software, version 4.0 (Blender Development Team, 2023; Lautenschlager, 2017). The medullary space was reconstructed based on the internal vertebral model recreated by Williams et al. (2021), which was based on the trabecular architecture of an azhdarchoid pterosaur. Initially, we connected the cranial and caudal ends of the neural canal with a hollow cylinder. Then, we added smaller cylinders with one mm in diameter to represent the trabeculae. These cylinders were organized in groups of 10 units arranged around and perpendicular to the neural canal in the medullary space of the postaxial cervical vertebrae, with the innermost end contacting the neural canal and the opposite end fixed to the internal wall of the vertebral cortex (Fig. 1B). In total, the number of cylinders representing the trabeculae varied from 20 to 150 units, depending on the length of the vertebra (Table 1). The trabeculae were not added to the atlas-axis model, as these vertebrae lack pneumatic foramina in Azhdarcho lancicollis (Averianov, 2010).

Table 1 Number of cylinders representing the trabeculae in the medullary space of the vertebrae of Azhdarcho lancicollis.

Vertebra	Cylinders	
Atlas + axis	0	
Cv III	70	
Cv IV	100	
Cv V	150	
Cv VI	100	
Cv VII	70	
Cv VIII	40	
Cv IX	20	
Notes.

Abbreviation Cv cervical vertebra

Roman numerals indicate the position of the vertebra in the cervical series.

The most vulnerable regions to receiving stress were identified using Finite Element Analysis (FEA), a non-invasive method that allows the discretization of continuous systems to observe element deformation (Rayfield, 2007; Kupczik, 2008; Wintrich et al., 2019; Button et al., 2023). The analysis was performed separately on each vertebral model. In the preprocessing stage of the analysis, we discretized the systems, assigned the material properties, and applied the muscle load vectors using Hypermesh software, version 13.0 (Rui & Jianmin, 2008). The number of recovered elements varied in the corresponding discrete systems of each model (Table 2).

Table 2 Number of elements in each model after preprocessing for Finite Element Analysis.

Vertebra	Anhanguera piscator	Azhdarcho lancicollis	
Atlas-axis	1,145,196	1,146,864	
Cv III	1,145,196	1,276,250	
Cv IV	1,614,805	1,246,029	
Cv V	1,236,992	1,204,577	
Cv VI	1,198,414	1,269,040	
Cv VII	1,441,387	1,259,287	
Cv VIII	1,622,939	1,227,213	
Cv IX	1,450,586	1,210,259	
Notes.

Abbreviation Cv cervical vertebra

Roman numerals indicate the position of the vertebra in the cervical series.

Each vertebral model was defined as a linear elastic material considering Young’s modulus E = 22 GPa and Poisson’s ratio v = 0.3, as previously used in pterosaur vertebrae (Williams et al., 2021). The materials were treated as homogeneous and isotropic, due to the difficulty in recognizing a possible anisotropy of the trabecular structures (Odgaard, 1997; Kabel et al., 1999; Herbst et al., 2021). Muscle loads were defined based on the “Maximal Force Production” (Fpmax), which is widely used to estimate the muscular capacity of extinct animals (Porro et al., 2011; Bishop, Cuff & Hutchinson, 2021) and is expressed through the following formula: (1) Fpmax=mmusc.σ.cosαoρ.lo

where mmusc corresponds to muscle mass, σ represents the maximum stress developed in the muscle fibers, cos(αo) is the cosine of the pennation angle in the fiber length, ρ is equivalent to muscle tissue density, and 𝔩o represents the ideal length of muscle bundle (Porro et al., 2011; Bishop, Cuff & Hutchinson, 2021; Buchmann & Rodrigues, 2025). The Fpmax used as muscle loads here were calculated from thirteen cervical muscles by Buchmann & Rodrigues (2025), and are listed in Table 3. The application of the load vectors in the discretized models was made using OpenSim software, version 4.4 (Delp et al., 2007) and followed the path of each muscle (Fig. 2), as previously determined by Buchmann & Rodrigues (2025) according to the locations of muscle attachments. The applied load was equally distributed across the vectors and along the predicted muscle path. The orientation of the vectors was based on the angle of each vertebra in relation to the cervical position at rest, as defined by Buchmann & Rodrigues (2024b) (Fig. 2), although each model was analyzed separately. The constraints were applied in the region opposite the direction of the load vector (Fig. 2) (Lautenschlager et al., 2016). We organized the muscles into six distinct scenarios to simulate head and neck movements during the life of both pterosaurs, representing rotations around the sagittal (yaw) and transverse (pitching) axes (Table 3; Fig. 2). Pitching was defined as dorsal and ventral, which is justified by the distinction of the muscles responsible for elevation and lowering of the skull and vertebrae (Table 3; Fig. 2). We determined the deformation of the materials with static analysis, which is based on asymmetric tension and compression deformation (Williams et al., 2021). All models discretized in Hypermesh are available online and can be accessed by the through the DOI provided as Declared Data.

Table 3 Muscles and their respective Fpmax in the six movement scenarios created for this analysis.

List of neck muscles and their Fpmax from Buchmann & Rodrigues (2025).

Cervical movement	Muscles	Anhanguera piscator
Fpmax (N)	Azhdarcho lancicollis
Fpmax (N)	
Dorsal pitching	Transversospinalis capitis	46.504	41.666	
(head)	Complexus	189.452	44.776	
	Splenius capitis	93.831	16.681	
Dorsal pitching	Transversospinalis cervicis	126.260	76.170	
(neck)	Intercristales	27.890	1.114	
	Interspinales	0.499	0.079	
Yaw (head)	Complexus	94.726	22.388	
	Longissimus capitis superficialis	43.535	19.305	
	Rectus capitis lateralis	18.243	13.318	
Yaw (neck)	Transversospinalis cervicis	63.130	38.085	
	Longissimus cervicis	20.353	8.175	
Ventral pitching	Rectus capitis ventralis	183.864	49.558	
(head)	Longissimus capitis profundus	37.898	12.362	
Ventral pitching	Longus colli	150.756	85.734	
(neck)	Flexor colli	29.672	10.168	
	Longissimus cervicis	40.706	16.350	

Figure 2 Locations where the load vectors and constraints were placed.

Schematic drawing representing the dorsal pitching of the head (A) and neck (B), yaw of the head (C) and neck (D), and ventral pitching of the head (E) and neck (F) of Anhanguera piscator. The positions of the load vectors are represented in the articulated cervical series and in the fifth vertebra in right lateral and cranial views, respectively; the positions of the constraints are represented only in the second.

The processing was performed in Abaqus software, version 6.14-1, with which von Mises stresses were calculated based on the simulated load on each vertebra during each analyzed scenario (Barbero, 2023). The calculations considered an average limit of 99%, which disregarded only the areas with stress higher than the pre-established properties (Montefeltro et al., 2020; Barbosa et al., 2023). We used contour plots to illustrate the contrast between regions with higher and lower von Mises stresses (Rayfield, 2007; Rayfield, 2012). In the contour plots, cooler colors represent minimal or absent stresses and warmer colors represent increasing stresses (Rayfield, 2007; Rayfield, 2012). We defined the maximum limit of the plots at 75 and 50 MPa for Anhanguera and Azhdarcho, respectively, which allows us to visualize the distribution of stresses that are lower than the determined values. The extrapolation of the pre-established limits for the plots was represented in gray, corresponding to a range between the maximum limit and the highest stresses found in each model (Rayfield, 2007; Rayfield, 2012). Finally, we generated averages of the von Mises stresses found in each element.

Results

The von Mises stresses generated by FEA in each model are available online and can be accessed through the DOI provided as Declared Data. In Anhanguera piscator, von Mises stresses averages were generally higher in vertebrae of the caudal half of the neck, except during ventral pitching of the head (Table 4; Fig. 3). However, this pattern is not repeated when we observe the maximum von Mises stresses of each vertebra (Table 4; Fig. 3).

Table 4 Average and maximum von Mises stresses (in MPa) per cervical vertebra during the six tested scenarios.

Cervical movement	Avg Anhanguera piscator	Max Anhanguera piscator	Avg Azhdarcho lancicollis	Max Azhdarcho lancicollis	
Atlas-axis					
Dorsal pitching (head)	0.1903628	70.608	0.4532706	42.622	
Dorsal pitching (neck)	0.1362716	146.882	0.698867	76.584	
Yaw (head)	0.06946402	68.527	0.1859707	28.796	
Yaw (neck)	0.0731491	146.876	0.4408102	73.660	
Ventral pitching (head)	0.1778092	58.656	0.2643937	30.977	
Ventral pitching (neck)	0.1260076	45.858	0.4321745	44.971	
Cv III					
Dorsal pitching (head)	0.1432122	66.628	0.1128823	15.019	
Dorsal pitching (neck)	0.07075787	15.672	0.08828958	11.498	
Yaw (head)	0.08090331	13.330	0.07944777	9.107	
Yaw (neck)	0.05077308	10.918	0.06753862	7.860	
Ventral pitching (head)	0.1230353	35.348	0.0941890	6.887	
Ventral pitching (neck)	0.1060403	17.686	0.1514276	12.733	
Cv IV					
Dorsal pitching (head)	0.1906578	89.946	0.08347269	8.493	
Dorsal pitching (neck)	0.1005121	32.743	0.07565448	8.025	
Yaw (head)	0.1251783	89.807	0.07294034	5.040	
Yaw (neck)	0.07017452	32.721	0.06334519	8.023	
Ventral pitching (head)	0.1703538	20.495	0.07159541	6.019	
Ventral pitching (neck)	0.1542806	22.389	0.134565	9.629	
Cv V					
Dorsal pitching (head)	0.2057354	23.280	0.1002856	15.957	
Dorsal pitching (neck)	0.1034172	8.979	0.03244875	2.539	
Yaw (head)	0.1327834	12.295	0.06658879	20.583	
Yaw (neck)	0.07264016	8.971	0.02656626	2.538	
Ventral pitching (head)	0.1699669	17.620	0.03541577	22.530	
Ventral pitching (neck)	0.140279	5.877	0.05936433	6.637	
Cv VI					
Dorsal pitching (head)	0.4787241	75.156	0.06570151	5.635	
Dorsal pitching (neck)	0.1055579	16.171	0.09406872	7.664	
Yaw (head)	0.2860401	22.825	0.0526079	7.644	
Yaw (neck)	0.07596325	16.162	0.07963328	6.434	
Ventral pitching (head)	0	0	0	0	
Ventral pitching (neck)	0.1794746	17.435	0.1765258	17.603	
Cv VII					
Dorsal pitching (head)	0.1132572	38.230	0.05849235	9.451	
Dorsal pitching (neck)	0.158348	68.668	0.07681324	8.704	
Yaw (head)	0.06432068	18.008	0.04286237	8.057	
Yaw (neck)	0.08042746	25.939	0.076084	8.682	
Ventral pitching (head)	0	0	0	0	
Ventral pitching (neck)	0.1878169	12.512	0.1196111	18.339	
Cv VIII					
Dorsal pitching (head)	0.2195108	26.281	0.07193244	11.251	
Dorsal pitching (neck)	0.3158771	26.015	0.08542023	14.329	
Yaw (head)	0.1599809	11.385	0.05237828	12.832	
Yaw (neck)	0.2216802	26.009	0.07523868	14.324	
Ventral pitching (head)	0	0	0	0	
Ventral pitching (neck)	0.787739	57.599	0.1323972	13.065	
Cv IX					
Dorsal pitching (head)	0.2553141	20.098	0.3735559	104.339	
Dorsal pitching (neck)	0.3272258	19.698	0.3200939	34.436	
Yaw (head)	0.2672971	32.379	0.1598384	18.070	
Yaw (neck)	0.2247806	21.875	0.3337474	34.269	
Ventral pitching (head)	0	0	0	0	
Ventral pitching (neck)	0.6225074	21.878	0.42996777	67.818	
Notes.

Abbreviation Avg average of the von Mises stresses

Cv cervical vertebra

Max maximum von Mises stress

Roman numerals indicate the position of the vertebra in the cervical series.

Figure 3 Graphs showing the von Mises stress of each vertebra of both pterosaurs.

The average and maximum von Mises stresses are shown on the top and bottom graphs, respectively. Abbreviation: ATAX, atlas-axis. Roman numerals indicate the position of the vertebrae within the cervical series. Stresses are presented in MPa.

Specifically in the cervical movement scenarios of Anhanguera piscator, the highest averages were recorded in the posterior cervicals (Table 4; Fig. 3). In this pterosaur, the average stresses during dorsal pitching of the neck were prominent in the eighth and ninth vertebrae, indicating that cervical pitching was the movement that most influenced the biomechanical behavior at the base of the neck (Figs. 3 and 4). If we disregard the three caudalmost cervical vertebrae, the highest von Mises averages were observed during head movements, except in the sixth vertebra during ventral pitching of the head (Table 4; Fig. 3). The increase in stress generated by head movements throughout the cervical series does not follow exactly the same pattern observed in neck movements. During dorsal pitching and head yaw, the highest average stresses were observed in the sixth cervical. Except for this vertebra in both head movements, there is an increase in von Mises averages from the cranial to the caudal end of the neck, as well as in cervical scenarios.

In contrast, the ventral pitching of the head in Anhanguera piscator demonstrates the highest von Mises average on the atlas-axis, as well as increased stresses in the cranial half of the neck (Table 4; Fig. 3). However, it should be noted that there is no musculature responsible for the ventral pitching of the head acting on the caudal half of the neck, which consequently does not generate stress (Table 4; Fig. 3). Complete head pitching (i.e., the dorsoventral movement) was responsible for the highest von Mises averages in the atlas-axis of Anhanguera piscator, indicating that movements through the lateral axis were those that most influenced the biomechanical behavior close to the skull (Table 4; Fig. 3). Neck yaw generally resulted in the lowest von Mises averages, except in the seventh vertebra, whose stress was lowest during yaw of the head (Fig. 2). The low stresses generated during yaw were mild on the lateral sides of the vertebrae, which is consistent with the presence of pneumatic foramina in biomechanically less affected regions (Fig. 4). However, the sixth cervical presented higher stresses in the pre-zygapophyses that may have influenced the biomechanical behavior of the entire lateral of the vertebra.

Figure 4 Von Mises stress contour plots of the cervical vertebrae of Anhanguera piscator.

Right lateral view. The interpretive drawings above the plots show the location where the load vectors were applied, with the color of the arrows following those of Fig. 2. Pneumatic foramina are in light gray. Abbreviation: ATAX, atlas-axis. Roman numerals indicate the position of the vertebrae within the cervical series. Stresses are presented in MPa. Models are not to scale.

The distribution of stresses was more accentuated in both cervical ends of Anhanguera piscator than in the mid-cervicals, propagating with greater intensity over the lateral foramina of the atlas-axis and posterior cervical vertebrae (Fig. 4). The pneumatic foramina located caudally to the bases of the transverse processes of the posterior cervical vertebrae also received higher stresses, mainly during cervical movement scenarios (Fig. 5).

Figure 5 Von Mises stress contour plots of the cervical vertebrae of Anhanguera piscator.

Caudal view. Roman numerals indicate the position of the vertebrae within the cervical series. Stresses are presented in MPa. Models are not to scale.

In Azhdarcho lancicollis, the von Mises averages and the maximum stresses followed the same pattern in each vertebra, being in both cases higher at the cervical ends. The atlas-axis had the highest average stresses in all scenarios, especially during dorsal pitching of the neck (Table 4; Fig. 3). The stresses generated by dorsal pitching of the head were also prominent at the cranial end of the neck, indicating that this movement affected this cervical region the most (Fig. 6). The von Mises averages of the other neck movements were also higher than those of the head in the atlas-axis, demonstrating that the cervical scenarios had a greater influence on the biomechanical behavior near the skull. The absence of pneumatic foramina in the atlas-axis cortex may be related to the accumulation of stresses in these vertebrae (Fig. 6).

Figure 6 Von Mises stress contour plots of the cervical vertebrae of Azhdarcho lancicollis.

Right lateral view. The interpretive drawings above the plots show the location where the load vectors were applied, with the color of the arrows following those of Fig. 2. Abbreviation: ATAX, atlas-axis. Roman numerals indicate the position of the vertebrae within the cervical series. Stresses are presented in MPa. Models are not to scale.

In the postaxial vertebrae of Azhdarcho lancicollis, the ninth vertebra exhibited the highest von Mises stresses in all three cervical movement scenarios and in dorsal pitching and yaw of the head (Table 4; Fig. 3). This pterosaur generally presented the highest stresses at both ends of the neck, which differs from Anhanguera piscator (Table 4; Fig. 3), in which the ventral pitching of the head did not exhibit stresses in the posterior cervical vertebrae. However, in Azhdarcho lancicollis, the large von Mises averages observed only in the cranial half of the neck are similar to those in other movement scenarios, demonstrating that the ventral pitching of the head cannot be considered an exception (Fig. 3).

The pneumatic foramina in the postaxial vertebrae of Azhdarcho lancicollis received mild stresses compared to the dorsoventral and lateral regions of the vertebral cortex, although exceptions were observed in the fifth, sixth, and ninth vertebrae (Fig. 7).

Figure 7 Von Mises stress contour plots of the cervical vertebrae of Azhdarcho lancicollis.

Cranial view. Interpretive drawings on the top highlight the position of pneumatic foramina (in light gray). Roman numerals indicate the position of the vertebrae within the cervical series. Stresses are presented in MPa. Models are not to scale.

Discussion

The divergences in the von Mises averages along the neck of the analyzed pterosaurs can be attributed to the differences in the robustness of the muscular attachments in the cervical vertebrae (Buchmann & Rodrigues, 2025). The more pronounced stresses at the cranial and caudal ends of the neck may be related to the robust insertions and origins of the cervical musculature in the axis and in the posterior cervical vertebrae, respectively (Zweers, Vanden-Berge & Koppendraier, 1987; Dzemski & Christian, 2007; Chamero et al., 2014; Boumans, Krings & Wagner, 2015). However, the muscles responsible for head movements were not inserted on the atlas-axis, but rather on the occipital region of the skull, while the other muscles had multi-headed origins that inserted on different vertebrae and diffused stress into a larger area (Snively & Russell, 2007; Böhmer et al., 2020; Buchmann & Rodrigues, 2024a; Buchmann & Rodrigues, 2025). On the other hand, distortions in the stress distribution were also observed, occurring due to variations in robustness within the same muscle (Buchmann & Rodrigues, 2025). The increased von Mises average in the sixth cervical during dorsal pitching and yaw of the head in Anhanguera piscator represents a different case, and can be attributed to the robust caudal most origin of the complexus muscle (Buchmann & Rodrigues, 2025). Additionally, the robustness of muscular attachments also explains the variation of maximum tensions in relation to the von Mises averages in Anhanguera piscator, demonstrating that the second is the safest way of interpreting the data.

The propagation of stresses along the cervical series agrees with the neck postures previously inferred for both pterosaurs (Buchmann & Rodrigues, 2024a). The larger stresses in the caudal part of the neck of Anhanguera piscator would not support a cervical posture more perpendicular to the trunk, due to the additional stresses to which the base of the neck would be subjected to (Furet et al., 2018; Fasquelle et al., 2019; Buchmann & Rodrigues, 2024a). On the other hand, the stresses were more pronounced at both ends in the cervical series of Azhdarcho lancicollis, which allowed for greater sinuosity in the middle of the neck (Averianov, 2013; Molnar et al., 2015).

The distribution of stresses at the cervical ends probably occurs due to the morphology of the vertebrae (Currey, 2002; Fastnacht, 2005). The shorter centra contributed to the spreading of stresses, being a morpho-functional specialization that confers stability to the detriment of a larger intervertebral space (Dzemski & Christian, 2007; Tambussi et al., 2012; Gutzwiller, Su & O’Connor, 2013; Boumans, Krings & Wagner, 2015; Molnar et al., 2015; Grytsyshina, Kuznetsov & Panyutina, 2016; Vidal et al., 2020; Buchmann & Rodrigues, 2024b). However, the taller neural arch of the atlas-axis and posterior cervicals favored a combination of opposing tensile and compressive efforts to counter the dorsoventrally received stress (Christian & Preuschoft, 1996; Tambussi et al., 2012; Molnar, Pierce & Hutchinson, 2014). Especially in Azhdarcho lancicollis, the elongated vertebral centra of the mid-cervicals may have contributed to stress retention (Padian et al., 2021).

In Anhanguera piscator, the taller neural spines of the mid-cervical vertebrae prevented dorsal stresses from spreading through the neural arch, limiting compressive stresses generated by dorsal pitching of the head to the neural spines (Fig. 4). On the other hand, the stresses received after dorsal neck pitching were widely distributed throughout the neural arch, due to its long and wide morphology (Fig. 4). Conversely, in Azhdarcho lancicollis, the stress resulting from the pitching of the skull and neck affected the entire dorsal portion of the mid-cervicals (Fig. 6), due to the extremely small neural spines (Dzemski & Christian, 2007; Buchmann & Rodrigues, 2025).

In both pterosaurs, the mechanical response of the mid-cervicals to lateral stress was likely optimized by the wide neural arches. In contrast, the atlas-axis and ninth vertebra of both pterosaurs and the eighth vertebra of Anhanguera piscator showed a wide distribution of lateral stresses favored by the narrow neural arch (Figs. 4 and 6). Especially in the posterior cervical vertebrae, the elongated transverse processes prevented stresses from being distributed further along the neural arch (Figs. 5 and 7).

Besides the cortical morphology, the trabecular architecture of the medullary space also influences the mechanical response to muscle stress (Wolff, 1892; Cubo & Casinos, 2000; Fastnacht, 2005; Wedel, 2005; Martin & Palmer, 2014; Moore, 2020; Buchmann et al., 2021; Williams et al., 2021). In Anhanguera piscator, the higher stresses in the posterior cervicals are consistent with the abundant trabecular bone in their vertebral centra (Buchmann et al., 2021), which would add more elastic stability and could contribute to the viscoelastic reaction (Currey, 2002; Fajardo, Hernandez & O’Connor, 2007; Williams et al., 2021). The complex architecture of the neural arch processes probably contributed to sustaining a slightly higher level of pneumatization than in the centrum (Moore, 2020; Buchmann et al., 2021). In azhdarchoids, such as Azhdarcho lancicollis, the helical distribution of the trabeculae may have contributed to the homogeneous propagation of stresses in the postaxial vertebrae (Williams et al., 2021), ensuring the integrity of the structure and increasing its resistance in response to stress.

In the vertebral cortex, we observed variation in the position, size, and number of pneumatic foramina, which have already been linked to biomechanical requirements (O’Connor, 2004; O’Connor, 2006). In the vertebrae of Anhanguera piscator, the loads received laterally were concentrated on the neural arch and ventro-laterally on the centrum, regions that border the pneumatic foramina. However, stresses were seen in the lateral pneumatic foramen in all scenarios, although milder than in the regions where the loads were received (Fig. 4).

In contrast, in the atlas-axis and posterior cervicals of Anhanguera piscator, vertebral morphology favors the spreading of stresses on their compact centra and may be linked to the reduced lateral pneumatic foramina in these regions (Buchmann, Avilla & Rodrigues, 2019; Moore, 2020). Thus, the presence of pneumatic foramina was associated with cortical surfaces less susceptible to stress in Anhanguera piscator. However, the presence of foramina contributes to compromising cortical integrity, favoring the propagation of stresses. The same occurred in the caudal region of the pedicle of the posterior cervicals of Anhanguera piscator, where stresses may have been higher than in the more cranial vertebrae due to the presence of foramina (Moore, 2020; Buchmann et al., 2021).

Similarly, in Azhdarcho lancicollis, the tubular shape of the mid-cervical vertebrae allows stresses to spread more easily, making the presence of lateral pneumatic foramina unfeasible (Williams et al., 2021). On the other hand, the stresses produced by muscular loads were more limited cranially and caudally to the postaxial vertebrae, indicating that pneumatic foramina were arranged in structurally less vulnerable regions (O’Connor, 2004; O’Connor, 2006; Buchmann & Rodrigues, 2019; Buchmann, Avilla & Rodrigues, 2019). However, our analysis disregards the shear stresses that occurred in response to the sliding of the zygapophyseal facets (Buchmann & Rodrigues, 2024a).

The method used here analyzes free-form objects based on a simplification of the models, which results in approximate solutions (Fastnacht et al., 2002). The von Mises stresses found do not approach the probable yield stress of the material, suggesting that the vertebral structure could withstand higher loads before permanent deformation (Rayfield et al., 2001). However, the scenarios analyzed only consider the effects of bone tension and compression, which are mechanical stresses that bones are less susceptible to failure, due to their high axial stiffness (Fastnacht, 2005). The addition of hypothetical movements that include bone shearing and torsion would likely cause stresses through which the structures could be closer to the limit of elastic deformation (Fastnacht, 2005).

Implications for cervical movements during foraging

The morphological specializations of anhanguerids and azhdarchids indicate that they had predatory habits (Bestwick et al., 2018). However, there are still uncertainties regarding the prey capture style employed by both (Kellner & Tomida, 2000; Humphries et al., 2007; Witton & Naish, 2008; Witton & Naish, 2013; Averianov, 2013; Naish & Witton, 2017; Bestwick et al., 2018; Henderson, 2018). In Anhanguera piscator, the hypothesis of capture through aerial predation is supported by the adequate wing proportions for dynamic flight (Witton & Habib, 2010), while skimming seems unlikely due to the high energy expenditure and the absence of mechanical requirements in the skull and cranial portion of the cervical series (Humphries et al., 2007).

For azhdarchids, the most accepted proposals suggest the consumption of foods that would not require much bite force, based on their slender muscles in the jaw and cervical series (Witton & Naish, 2008; Witton & Naish, 2013; Averianov, 2013; Naish & Witton, 2017). Therefore, Azhdarcho probably fed on small prey, although supplementation with fruits and carrion is not ruled out (Prieto, 1998; Carroll, Poust & Varricchio, 2013; Witton & Naish, 2008; Witton & Naish, 2013; Naish & Witton, 2017).

In both pterosaurs, robust cervical musculature and intervertebral ligaments allowed rapid lunges of the head and neck, consistent with active prey capture in aquatic or terrestrial environments (Habib, 2015; Bestwick et al., 2018; Buchmann & Rodrigues, 2024a; Buchmann & Rodrigues, 2025). The mild distribution of stresses in mid-cervicals across all movement scenarios suggests that agile movements could have been performed without putting bone integrity at risk. Furthermore, the segmented ligaments and the ligamentum nuchae may have contributed to repositioning the neck after ventral pitching and keeping the skull elevated in relation to the trunk during foraging, incurring a lower energetic cost (Witton & Naish, 2008; Habib, 2015; Buchmann & Rodrigues, 2024a).

However, the concentration of stresses near the base of the neck in Anhanguera piscator and at both cervical ends in Azhdarcho lancicollis highlights how stability varied among pterosaurs, as also observed in other long-necked archosaurs (Dzemski & Christian, 2007; Taylor, Wedel & Naish, 2009; Guinard et al., 2010; Krings et al., 2014; Krings et al., 2017; Grytsyshina, Kuznetsov & Panyutina, 2016; Böhmer et al., 2020). Variations in stress accumulation also indicate that head and neck movements along the same axis can generate distinct biomechanical behaviors in both taxa, corroborating the hypotheses of different predation habits in these pterosaurs.

In Anhanguera piscator, the higher stresses exhibited at the base of the neck are consistent with a less vertical cervical posture in relation to the trunk, making the head closer to the water surface, as seen in birds that forage in shallow waters (Sick, 1997). Thus, predation on the banks or on the water surface seems to be the most biomechanically likely scenario for Anhanguera piscator (Habib, 2015; Buchmann & Rodrigues, 2025). Furthermore, the stresses concentrated on the neural spines during dorsal pitching of the head favored the integrity of the neural arch to the detriment of the generated stress, which enabled rapid head retractions (Habib, 2015; Pêgas, Costa & Kellner, 2020; Buchmann & Rodrigues, 2025).

Conversely, in Azhdarcho lancicollis the neck was more verticalized and had higher stresses on both ends, being compatible with terrestrial habits that require raising the head during swallowing (Witton & Naish, 2008; Witton & Naish, 2013; Naish & Witton, 2017). Additionally, reducing stresses in the long mid-cervical vertebrae could favor attack dexterity, as the sinuosity between the cranial and caudal third of the neck creates tensional integrity (Furet et al., 2018; Fasquelle et al., 2019; Buchmann & Rodrigues, 2024a). The tubular morphology of the mid-cervicals allows a wide distribution of stresses, being consistent with the execution of slow and longer-lasting movements, which could occur after pulling food when removing portions of a prey (Bowman et al., 1994; Witton & Naish, 2008; Naish & Witton, 2017; Buchmann & Rodrigues, 2025).

The models analyzed were subjected to the inferred maximum loads of the cervical musculature, which may or may not represent their actual habits in life (Jones, Brocklehurst & Pierce, 2021). We must consider that the three scenarios expressed by the movements of the head are complementary to those of the neck, increasing the intensity of the stresses in the vertebral series. Additionally, our limited prior knowledge about pterosaur foraging means that the tested scenarios may not correspond to actual cervical movements produced during foraging, although the numerical results are correct (Lauder, 1995; Fastnacht, 2005). Thus, our hypotheses regarding food capture methods were raised considering ground foraging, excluding the influence of drag on aerial locomotion (Rahman, 2017). We did not include a scenario of head rotation because pterosaurs probably did not perform this movement on land to capture food (Fastnacht, 2005), although skull torsion could have been performed to decrease the impact of drag after prey capture during flight (Henderson, 2018). The vertebral models also differ from the cervical arrangement of the animal in life, in which the vertebrae would be articulated through cartilage in the joints and ligaments (Dimery, Alexander & Deyst, 1985; Gál, 1993; Ponseti, 1995; Tsuihiji, 2004; Cobley, Rayfield & Barrett, 2013; Buchmann & Rodrigues, 2024b). However, shear stresses would be restricted to the joints and would not interfere with the stress generated by the muscular load (Teng & Herring, 1998; Fastnacht, 2005; Buchmann & Rodrigues, 2024a).

Conclusions

Our analysis reveals that, in both analyzed pterosaurs, stresses accumulated at the ends of the neck, with the biomechanical behavior more affected near the base of the neck in Anhanguera piscator and next to the skull in Azhdarcho lancicollis. However, there is an important difference in the distribution of stresses along the cervical series in both, with an increase in stresses towards the trunk in Anhanguera piscator, while the stresses were concentrated at both ends of the neck in Azhdarcho lancicollis. The tall neural arch of the vertebrae at the cervical ends allowed tension and compression forces to act in favor of controlling stresses, mainly those generated by the pitching of the head and neck. Additionally, the larger stresses are consistent with the internal trabecular reinforcement of these vertebrae, indicating that bone distribution in the medullary space is linked to biomechanical requirements.

Most pneumatic foramina were located in cortical areas that received less stress, indicating that the mechanical behavior influenced their position. Furthermore, pneumatic foramina are notably smaller and fewer on vertebrae that presented higher stresses, suggesting that biomechanical requirements also interfere with the size and number of foramina.

The compressive forces were consistent with different predation habits for each pterosaur. The differing accumulation of stress indicates a more horizontal posture of the neck relative to the trunk in Anhanguera piscator and a more vertical posture in Azhdarcho lancicollis, which are consistent with the predation habits on the water margins and on land, respectively, as previously hypothesized.

The great capacity of the neck to absorb stresses after ventral pitching of the head and neck indicates these movements could be performed rapidly in both pterosaurs, as expected in predatory habits. However, anatomical differences in the vertebrae favored variations in stress concentration during dorsal pitching of the head and neck, which indicated a probable faster elevation of the skull and neck in Anhanguera piscator, and stronger in Azhdarcho lancicollis.

We thank Takanobu Tsuihiji, Makoto Manabe, and Chisako Sakata from the National Museum of Nature and Science, Japan, for providing us with the CT scans of the vertebrae of Anhanguera piscator and Alexander Averianov from the Zoological Institute of the Russian Academy of Sciences for providing us with the digitized models of the vertebrae of Azhdarcho lancicollis. We thank Felipe Montefeltro and Gabriel Gonzalez for sharing their knowledge about the Finite Element Analysis method and the use of the software.

Additional Information and Declarations

Competing Interests

Author Contributions

Data Availability

The authors declare there are no competing interests.

Richard Buchmann conceived and designed the experiments, performed the experiments, analyzed the data, prepared figures and/or tables, authored or reviewed drafts of the article, and approved the final draft.

Taissa Rodrigues conceived and designed the experiments, authored or reviewed drafts of the article, and approved the final draft.

The following information was supplied regarding data availability:

The data and models are available at Figshare: Buchmann, Richard (2025). Models and von Mises stress of the cervical vertebrae of Anhanguera piscator and Azhdarcho lancicollis. figshare. Dataset. https://doi.org/10.6084/m9.figshare.30260125.v1.

The three-dimensional models are generated from tomograms of the vertebrae of NSM-PV 19892 (Anhanguera piscator) and from scanning of specimens ZIN PH 105/44, ZIN PH 131/44, ZIN PH 144/44, CCMGE 1/11915, ZIN PH 139/44, ZIN PH 147/44, ZIN PH 138/44, ZIN PH 137/44, ZIN PH 122/44, and CCGME 7/11915, attributed to Azhdarcho lancicollis.

The models containing the load vectors and constraints, along with the von Mises stresses found in each analysis, are also available.

The three-dimensional models generated from the vertebrae of Anhanguera piscator are available at Morphosource:

- dx.doi.org/10.17602/M2/M589278 (Media ID 000589278)

- dx.doi.org/10.17602/M2/M589282 (Media ID 000589282)

- dx.doi.org/10.17602/M2/M589286 (Media ID 000589286)

- dx.doi.org/10.17602/M2/M589290 (Media ID 000589290)

- dx.doi.org/10.17602/M2/M589294 (Media ID 000589294)

- dx.doi.org/10.17602/M2/M589298 (Media ID 000589298)

The three-dimensional models generated from the vertebrae of Azhdarcho lancicollis are available at Morphosource:

- dx.doi.org/10.17602/M2/M568148 (Media ID 000568148)

- dx.doi.org/10.17602/M2/M568153 (Media ID 000568153)

- dx.doi.org/10.17602/M2/M568158 (Media ID 000568158)

- dx.doi.org/10.17602/M2/M568163 (Media ID 000568163)

- dx.doi.org/10.17602/M2/M568168 (Media ID 000568168)

- dx.doi.org/10.17602/M2/M568173 (Media ID 000568173)

- dx.doi.org/10.17602/M2/M568178 (Media ID 000568178)

- dx.doi.org/10.17602/M2/M568183 (Media ID 000568183)

- dx.doi.org/10.17602/M2/M568188 (Media ID 000568188).

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
