# Peer review of "Resistance of cervical vertebrae in response to muscular stresses in pterosaurs: implications for foraging habits and skeletal pneumatization"

_PeerJ, doi:10.7717/peerj.20388_

## Round 0.1 · original submission · Major Revisions

· Academic Editor

Major Revisions

The two referees make similar overall comments that are generally positive, but that the manuscript generally needs more details about the exact processes of the models and analyses in order to be clear about what was done and therefore the interpretation of the results.

·

Basic reporting

The overall reporting is largely OK but could do with more clarity. In particular the introduction can be improved in structure. There are a lot of themes/topics packed into the introduction but a common thread and logical flow is missing. This can be improved by clearly stating the aims/hypotheses and what is actually tested (this is in there but a bit difficult to tease apart). For example, make it clear what assumptions are being made with regards to the different species and their feeding styles, how this is tested and what kind of biomechanical perfomance metrics you would expect.
For example, in line 110 "distinct feeding" habits are mentioned but never explained which ones these are until a lot later in the discussion. I am sure the authors know all of this information but the readers may not be experts so please make this clearer and provide more detail even if you think it is self-explanatory.

Experimental design

The idea behind this study is really cool and I looked forward to seeing how FEA could potentially inform on the different neck postures, feeding styles, etc. Further, it is nice to see FEA applied to a different region than just the skull. Unfortunately, the execution of this is lacking. Not necessarily in the way the analyses were performed but how everything is presented. However, due to the lack of key information I cannot tell whether one or the other are the case.

Importantly, the details for the FEA are not sufficiently explained:
- How were the models constrained? At which points/nodes, how many, differently for different scenarios, etc.?
- Muscle loads are unclear. How where the muscle forces calculated, where were they applied to, are there different muscles/muscle vectors for different scenarios?
- As I understand it, the individual vertebra are not connected for the analyses; this is fine but have they been orientated in an in-vivo fashion or just been place horizontally (in the introduction and elsewhere the sigmoidal morphology of the necks are mentioned, so was this considered)?
- The terms loads, stress and strain appear to be used randomly, but all of these are different things with very concrete meanings and cannot be used interchangeably. Loads are the forces applied to an FEA model (e.g. muscle loads) - these are not stresses but will result in stresses. Stress and strain are different metrics of functional responses/metrics in a geometry due to the external forces but these are distinct as well. If you show von Mises stress then you only have stresses not strains. For the latter you will need to display/ measure them separately. On a similar note, the term cortex is used throughout the text but it is unclear what this means in this context? The surface of the model, cortical bone, etc.? And why would stresses only be transferred along the cortex and not the entire model?
- There are six scenarios tested but not a single one is explained in terms of the analysis setup.

In addition to the points above, there needs to be a figure which shows the neck models with muscle attachments mapped, possible muscle vectors/directions shown, ideally the different scenarios visualised in some way


Table 4 lists the average von Mises values per model. Average/mean values can be tricky as very disparate stress responses may result in the same mean value. However, there are not necessarily other metrics that can be used easily. Some examples for alternatives are below if interested, Regardless of which metric and style are used, it would be very helpful to plot the data in some form. Table 4 is not very intuitive and digestible so presenting the data in some graphical form would bring the message across a lot better.

Marcé-Nogué, J., De Esteban-Trivigno, S., Püschel, T. A., & Fortuny, J. (2017). The intervals method: a new approach to analyse finite element outputs using multivariate statistics. PeerJ, 5, e3793.

Tumelty, M., & Lautenschlager, S. (2025). Is cranial anatomy indicative of fossoriality? A case study of the mammaliaform Hadrocodium wui. The Anatomical Record.

Meade, L. E., Pittman, M., Balanoff, A., & Lautenschlager, S. (2024). Cranial functional specialisation for strength precedes morphological evolution in Oviraptorosauria. Communications biology, 7(1), 436.

Lautenschlager, S., Fagan, M. J., Luo, Z. X., Bird, C. M., Gill, P., & Rayfield, E. J. (2023). Functional reorganisation of the cranial skeleton during the cynodont–mammaliaform transition. Communications Biology, 6(1), 367.

Validity of the findings

Difficult to say something about this given the sparse information regarding the methods and the different scenarios. I am happy to look at a revised version but cannot make any fair judgement of this at this stage.

Additional comments

There are a lot of good things in this study/manuscript but it is not fully there yet. There needs to be more information regarding the hypotheses and methodological setup to make this a really good and interesting paper.

·

Basic reporting

Buchmann and Rodrigues have used FEA modelling of the cervical vertebrae series of two pterosaur species under stresses from the muscles to examine two related areas. Firstly, to compare areas of stress with the location of the foramen for skeletal pneumatization with the resultant stress fields in order to determine how structural constraints affect pneumatization morphology. Secondly, the authors link the stresses experienced through the vertebrae with potential implications for cervical and cranial orientation while foraging. Overall, the research is interesting and deserves to be published, however it does require some clarification work to reach this level.
With regards to basic reporting, beyond some minor typographic errors, there are multiple areas that could benefit from rewriting to improve clarity.
The biggest issue that needs to be addressed is the lack of any figure showing the proposed orientation scenarios of the skull and cervical vertebrae being tested. As such it is unclear where the forces are being applied and in what directions, as well as what degree of flexion is being considered. As part of this it is also necessary to show how the muscles are attached, even just a generalized diagram showing the lines of action would help though ideally for FEA it would be better to show the selected vertices that the forces were applied to, though that sort of detail may be better in the supplement than the main text. Since the analyses were conducted on the different vertebrae independently it might be clearer to have a two part figure with part one showing a generalized image showing the entire cervical series and skull with lines of action for the tested muscles and a second part with arrows or transparent variants indicating the different flexion criteria (dorsal/ventral/lateral flexion of the head, and dorsal/ventral/lateral flexion of the neck). Then in the subsequent figures (2-6) including arrows showing where the forces were applied in the experimental setup on the individual vertebrae so the reader can follow the direction the forces are acting in. This will be especially important for the clarity of the ventral flexion of the head FEA models since these elements have not had any applied at all as none of the selected muscles attach to them.
These additional clarifications in the figures would also assist in the discussion. A figure showing the myology or even assumed lines of action would also be helpful when the muscular insertions are considered in the discussion (Lines 266 - 273) while a figure showing the orientation scenarios would help a reader understand what the unsupported “cervical posture more perpendicular to the trunk” (lines 275-277) actually entails.
The introduction does well at framing the background and upcoming analyses, but clarity could be improved by adding to or reworking the final paragraph of the introduction to more clearly state what the authors are proposing they test. As it currently reads the authors are simply stating that these things could be tested not that they are going to in any meaningful way.
Structurally the manuscript as presented is excellent though I would recommend switching the order of figures 4 and 5 so that all of the Anhanguera FEA model views are together. As it stands figure 4 is not cited at all in the text however, I suspect this is simply a typographic oversight since there are multiple places where it could be linked.
Other than the requested inclusion and modification of figures above there is also an issue with transposing of figures for consistency. The current figures 2-6 are inconsistent in their layout with figures 2 and 6 having the tested orientation (e.g. dorsal flexion of the head) along the top with the vertebral sequence in columns beneath. Figures 2-5 are transposed so the tested orientations are along the side, and the cervical series are in rows. Further the order of the cervical series within the row changes so that figure 4 is opposite to 3 and 5 because of the right lateral images used. Overall, this becomes very confusing to the reader. I would recommend following a consistent orientation across all the figures even though not all of them include the line drawings showing the foramen.
Throughout the results and discussion (first appearance line 214), the authors switch to describing dorsal, ventral, and lateral flexion of the head as pitching and yawing though not in all instances (e.g. line 251 reverts to flexions). While not incorrect as descriptions there is a loss of clarity in changing terminology from what is used in the methods, figures and tables. This could be easily addressed by changing the terminology to be consistent throughout by changing all discussion of head orientation scenarios to use pitch and yaw (though this would require specification of pitching up and down) so the head and neck scenarios are different or making everything consistent and just using flexions.

Experimental design

The experimental design appears to be logical however there are some parts that require further clarification before I can confidently say that the experiment has been described sufficiently let alone is replicable.
The primary thing that requires clarification is the orientation and application of the muscle loads. This has already been covered above when discussing figure clarity, but it is important to point out here that it is hard to determine if the experiment makes sense without seeing how the muscle force was applied. This issue is even more prevalent at lines 188 -190 which are severely limited by the lack of any presentation of what the proposed head and neck movements look like. I agree with the selection of muscles stated in Table 3 but without indications of where the forces are applied it is practically impossible to determine if the modelling is actually showing what the authors say it is.
I would like to commend the authors for making the models of the different elements available on Morphosource, but I would like clarification on whether these models are the base models or the discretized versions that include the cortex and trabeculae modelling and with have been remeshed for FEA analysis matching Table 2. As both the trabeculae modelling and discretizing process can influence the result it would be good to know whether the finalised FEA mesh is available.
Further I would be interested to know why the scales are inconsistent at the top end of the scales in the FEA figures. Anhanguera von Mises stress caps out at 70MPa while Azhdarcho caps out at 40MPa. Both are significantly higher than your reported max value seems to be 0.78MPa so I am unclear why they would need to be different as the cap of the previous bin in both is 0.25MPa. It may be purely an aesthetic choice, but it does mean a lot of the reported values are lumped into a very large grey bin, especially for the Atlas-Axis and CVIII/CIX.

Validity of the findings

Without the context provided by knowing where the muscle forces were applied to the meshes it is hard to be certain how sound the results are. Assuming that the muscle force applications are logical, and the issues with stress and strain are clarified (see below) then the conclusions, especially related to the implications in foraging are very well presented.
Throughout the results and into the discussion the authors discuss both stress and strain values while directing the reader to Table 4 however as far as I can tell Table 4 is just average von Mises stress values. There is no information regarding deformation caused by the stress (aka strain) on the elements that I can find which makes the switching between stress and strain in the results confusing. I struggled especially to follow the statements regarding the stress contour plots and location of the foramina especially the sentence from line 248-250 (“in the mid-cervical vertebrae…”). I feel that sentence in particular may need clarification. As I understood it the authors claim that the strain is lower near the foramen in cervicals IV-VII in the dorsal and ventral flexion of the head scenarios despite the figure cited (fig 2) showing moderate stress in those areas relative to the rest of the bones. I am not entirely sure where the “fewer strains” is drawn from or how they are getting that information from a table and figures showing stress. Potentially the authors are assuming the stress strain relationship at these values are linear but at by that logic I do not understand how the strain can be lower at this point.
When combined with the lack of a figure showing the directions of the forces applied it makes the comments about compressive stress in Anhanguera vs strain distribution in Azhdarcho for the neural spines during discussion (lines 298-304) more difficult to follow and therefore less impactful as well. Assuming it is all meant to be stress than the discussion does seem to make sense though.
As a minor side comment, I am a little surprised that the supplement Abaquus output files are in .txt format. This is not a bad thing, theoretically the data is easier to read that way and it seems to all be there so that is not an issue, just surprising.

Additional comments

Lines 50 – 53 “The convergences between birds…” not strictly incorrect but it might be good to include something like Witton & Habib 2010 (doi:10.1371/journal.pone.0013982) or similar to highlight the risks of using birds as analogues for pterosaurs as currently the nuance is unclear
Lines 65 – 70: “Analyses of the functional morphology…” these two sentences are largely saying similar things and can probably be combined to improve the flow of the paragraph
Lines 72 – 77: “The biomechanical behaviour of a bone…” I recommend splitting this into two sentences, probably by adding a “.” after the first set of references and removing the “and” so that it flows better. Currently it is a very long sentence
Lines 77 – 81: “Inferences of the possible movements…” I am not entirely sure what this sentence adds to the paragraph after talking about mechanical response. Likely it just needs something to tie it in but as it stands it is a weird jump in the flow of the paragraph
Line 206: “greatest strains at the neck ends” This is a confusing way of saying what is more clearly stated in the following two sentences. I would probably recommend just removing that sentence and just having the following two.
Line 236: “generated high strains” is an interesting statement when your largest strain is >1MPa. I assume it is relative to the rest of the modelling, but I would be interested to see some sort of commentary at some point reflecting on the very low values involved. As it stands even the highest strains reported here are not even close to what is commonly considered a failure point for bone (I was under the assumption that most studies consider 150 MPa as bone failure, though with safety factors I could understand the 70MPa that is the max on your scales for Figures 2,3, and 5)
The Implications for cervical movements during foraging section (Line 361 – 439) is excellent
Figure 1: I would recommend adding some sort of arrow or colouring to highlight the foramen as it is not especially clear as presented. I would also recommend adding an insert to B showing the internal structure that the model is simulating, potentially 2A from the Williams paper or something.
Table 2: this is good reporting but not as useful as the actual models depending if you shared the FEA models or not
Table 3: as stated previously these muscles and fmax values are nice but not useful without stating where they are acting on the FEA models
Lines 281 -282: “less resistance against structural failures” as mentioned previously while technically true the stress values presented here are very low and unlikely to approach what would generally be treated as structural failure so a comment on this in the discussion might be nice.
Lines 294 – 295: I would move the Padian reference earlier in the sentence, probably immediately before “especially” to avoid potential confusion. As it stands a reader could assume that Padian made statements about the strain in Azhdarcho in his Quetzalcoatlus biomechanics paper.
Lines 318 – 321: this is more a curiosity question than something that needs to be addressed but since the helical trabecular structure is one that was modelled for this study would it be worth also running a model with a different structure to see how much of an effect the helical structure that was modelled actually had? Also was there any sensitivity testing on how the number of trabeculae modelled effected the results at all?
Figure 4: this is not mentioned in the main text at all as far as I can tell but could fit easily into the Azhdarcho results section and at the end of line 304 in the discussion
Minor typographic issues
Line 74: “stresses on the its cortex” remove the additional “the”
Line 76: “at what point the elastic reactions tend to yield until the structure fractures” an awkward read, I suspect it should be “and the structure fractures” since once the elastic deformation yields the structure will deform permanently
Line 219: “fluence” possibly supposed to be influence but that would not fully make sense either
Figure 1: there is a space in the word cross-sections between the c and the r
Personal Preferences
These do not strictly need correcting, they are just things that were jarring to read
Line 51: “the first to the latter” – usually I expect the pairing of “former to latter” or “first to second” not a mix of both
Line 256: “strain fluency” I have never seen the word used this way, it might not be wrong but was strange to read
Line 344-347: “Our analyses did not…” This is great though if this is based on the analyses why have all the references? The way it reads currently makes your analyses seem less important. You can still have the references there but make sure the importance of your own analysis clearer by adding something like "in agreement with findings of previous studies" then the references or having that as a sentence after it.

---

## Round 0.2 · accepted · Accept

· Academic Editor

Accept

The referee is very happy with the changes, so I'm happy to accept the work.

·

Basic reporting

The work done by the authors has greatly improved the reporting. All of my previous comments and concerns have been thoroughly addressed, so much so that I don't have any further comments to add. The figure reworks especially are better than I could have hoped for.

Experimental design

The research is well defined and thorough. The additional figures and explanations in the methods helped clarify the approach and subsequent results beautifully.

Validity of the findings

I am able to open all of the supplemental material. The conclusions are very well presented as are the potential limitations.

Additional comments

I feel a bit bad not having much to say but the authors did excellent work addressing my concerns following my initial (and possibly overly extensive) review.